# Effect of *Tetragonia tetragonoides* (Pall.) Kuntze Extract on Andropause Symptoms

**DOI:** 10.3390/nu14214572

**Published:** 2022-10-31

**Authors:** Ka Youn Lee, Seung-Hyung Kim, Won-Kyung Yang, Geung-Joo Lee

**Affiliations:** 1Department of Horticulture, Chungnam National University, 99 Daehak-ro, Yuseong-gu, Daejeon 34134, Korea; 2Institute of Traditional Medicine and Bioscience, Daejeon University, 62 Daehak-ro, Dong-gu, Daejeon 34520, Korea; 3Department of Smart Agriculture Systems, Chungnam National University, Daejeon 34134, Korea

**Keywords:** *Tetragonia tetragonoides* (pall.) Kuntze, New Zealand spinach, halophyte, andropause symptoms

## Abstract

Testosterone and free testosterone levels decrease in men as they age, consequently inducing andropause symptoms, such as weight gain, fatigue, and depression. Therefore, this study aimed to evaluate the reducing effect of New Zealand spinach (NZS) on these androgenic symptoms by orally administering its extract to 26-week-old rats for four weeks. Biochemical blood testing was conducted, and the andropause symptoms-related indicators and muscular endurance levels were examined. In the NZS extract-treated rats, the decrease in muscle mass was suppressed, and immobility time was reduced in the forced swim test. In addition, the grip force and muscular endurance of the forelimbs were significantly increased compared to the control group; therefore, NZS extract exhibits a positive effect on the maintenance of muscle mass and improves muscular endurance. The representative male hormones, testosterone and progesterone, in the NZS extract-treated group were 1.84 times and 2.48 times higher than those in the control groups, respectively. Moreover, cholesterol and low-density lipoprotein, which affect lipid metabolism, were significantly reduced in the NZS extract-treated group. Overall, NZS extract shows potential for further development as a functional food material for improving muscle strength and relieving andropause symptoms.

## 1. Introduction

Decreased androgen levels induce andropause symptoms and reduce testosterone levels by 1–2% yearly [1]. The synthesis of testosterone is produced in the mitochondria through several stages of steroid hormone production from cholesterol as a raw material. Cholesterol is converted to pregnenolone by P450scc and StAR proteins, which is converted to progesterone by 3β-HSD2. CYP17A1 converts progesterone to androstenedione, and 17β-HSD3 synthesizes it into testosterone [2]. However, with aging, the function of the Leydig cells decreases and the expression of P450scc and StAR proteins decreases. In conclusion, as testosterone levels decrease, andropause symptoms appear [3]. On the other hand, testosterone is converted to DHT by 5α-reductase and to estradiol by aromatase, finally leading to a decrease in the total testosterone concentration [4]. The main andropause symptoms include sexual and erectile dysfunction, weight gain, muscle weakness, insomnia, fatigue, and depression [5,6]. Hormone and hormone-replacement therapies are prescribed for people diagnosed with androgen deficiency; however, some studies have suggested their association with certain risks, including prostate cancer and prostatitis [7,8]. Therefore, further research is required to explore potential natural sources to increase testosterone levels and relieve andropause symptoms.

The recent increase in andropause awareness has yielded a simultaneous upsurge in research on natural functional compounds for reducing andropause symptoms.

*Tetragonia tetragonoides* (Pall.) Kuntze is a halophyte plant of the Aizoaceae family. This plant, widely used by the M$\bar{a}$ori, was called K$\bar{o}$Kihi, and is now commonly known as New Zealand spinach. It is native to sandy soil on the seashore or coastal cliffs in New Zealand, Australia, Japan, and Korea [9]. Halophytes are known to maintain growth rates in salinity conditions through osmotic pressure control [10]. Plants tend to have a lower water potential to overcome the external water potential lowered by salinity. Therefore, the plant accumulates organic or inorganic solutes inside and increases the succulence [11]. Giulia Atzori et al. (2020) reported for NZS that when the seawater was treated, the leaf area decreased and succulence increased in the seawater treatment zone, and the concentrations of Mg (1.9 times), Cu (1.4 times) and Na (5.5 times) were increased compared to the control [12]. Therefore, NZS grown in sandy soil sprayed with seawater has a thicker mesophyll and higher sodium concentration than *Spinacia oleracea*, which is thought to be because, like the salt-tolerant crops reported by Glenn et al. (1999), it accumulates sodium in the vacuole to control the osmotic pressure [13]. However, high sodium concentrations in foods are associated with health risks. Caparrotta et al. (2019) confirmed that the sodium content in the leaves of *Spinacia oleracea* was significantly increased compared to the control when seawater was treated for 4 weeks [14]. However, the sodium content decreased through the cooking process of steaming or boiling, and this method is thought to be applicable to NZS. Meanwhile, Jaworska (2005) reported that the oxalate content of NZS leaves was 506–981 mg/100 g fresh matter, which was higher than that of other crops [15]. However, the water-soluble oxalate content constitutes 69–98% of the total oxalate content, so it is thought that the intake can be reduced by boiling or steaming.

Moreover, NZS has medicinal uses for gastrointestinal diseases such as gastric cancer, gastritis, gastric ulcer, and acid excess [16]. NZS extract contains various antioxidative compounds and exhibits antioxidant, antidiabetic, and anti-inflammatory effects [17,18]. Lee et al. (2019) isolated eight types of 6-methoxyflavonols from the aerial parts of the NZS [19], which all exhibited anti-inflammatory effects by inhibiting the production of pro-inflammatory mediators, such as inflammation-associated prostaglandins (PGE) and nitric oxide (NO), and showed antioxidant activity. Moreover, Choi et al. (2020) isolated 20 compounds from the aerial parts of NZS, which were expected to promote anti-inflammatory and antioxidant properties because they contained a catechol structure [20]. The administration of NZS extract to high-fat diet (HFD)-induced obese mice inferred anti-obesity effects by inducing a decreased adipogenesis-related gene expression, which suppressed the accumulation of total white adipose tissue and decreased the size of the adipocytes [21]. In addition, the administration of NZS extract to HFD-fed ovariectomized mice, to alleviate postmenopausal symptoms, lowered their body fat mass and improved their glucose tolerance; NZS extract also improved menopausal symptoms by lowering the concentration of pro-inflammatory cytokines, such as tumor necrosis factor-α (TNF-α) and monocyte chemoattractant protein-1 (MCP-1) [22]. However, although NZS extract has excellent lipid metabolism control and anti-inflammatory and antioxidant properties, studies on its effectuality on andropause are yet to be conducted. Therefore, this study evaluated the effect of NZS extract on alleviating andropause symptoms.

## 2. Materials and Methods

### 2.1. Extract Preparation from NZS Leaves

*Tetragonia tetragonoides* (Pall.) Kuntze (NZS) leaves were collected from the ocean dunes of Shinan-gun, Jeonnam Province, Korea. The aerial part of the NZS (500 g) was obtained through hot-water extraction using 10 L of 30% alcohol at 80 °C for 2.5 h. The filtered extracts were immediately treated to remove the ethanol in a rotary evaporator (N-1200 A, Eyela, Tokyo, Japan) under reduced pressure at 45 °C. The remnants were then frozen and freeze-dried in a dryer (Scanvac Coolsafe 95-15, Labogene, Lynge, Denmark) for 48 h, starting at a temperature of −95 °C under low pressure, which finally yielded a concentration of 11.2% [23].

### 2.2. Animal Study

Specific pathogen-free 26-week-old Sprague Dawley (SD) male rats were purchased from Daehan Biolink Co. (Eumsung, Korea). Each rat was housed in a cage of 28 cm × 42 cm × 19 cm with dates and bedding, and kept under a controlled temperature of 23 ± 2 °C and humidity of 55 ± 15% on a 12 h light/dark cycle by lighting at 9 a.m. A total of 15 SD rats were divided into three groups, which were treated as the negative control (Group 1), positive control (Group 2), and NZS (Group 3). Five rats as biological replications were used per test group. After four weeks of acclimatization, 300 mg/kg body weight of the complex extract was orally administered at 9 a.m. daily for four weeks. The control group was administered physiological saline containing 0.9 g/100 mL as a negative control group (Cleancle, JW Pharmaceutical, Seoul, Korea) and 300 mg/kg body weight of *Trigonella foenum-graecum* seed extract (Testofen^®^, Novarex, Cheongju, Korea) (positive control group), which has been reported as effective in relieving androgen menopausal symptoms [24].

### 2.3. Measure of Muscular Endurance

#### 2.3.1. Grip Force Test

To measure muscle endurance, the grip force (maximum force and temporal resistance) of the forelimbs or hindlimbs was measured three weeks after administration [25]. The maximum force of the forelimbs was automatically measured using a grip strength meter (47200, Ugo Basile S.R.L, Gemonio, Italy) [26].

#### 2.3.2. Forced Swim Test

The forced swim test (FST) was performed to measure muscular endurance three weeks after administration. The rats were placed in a glass cylinder (height, 50 cm; diameter, 20 cm) filled with water at 25–27 °C and forced to swim for 12 min [27]. Passive immobility and active behavior time was measured and analyzed with a smart v3.0 video tracking system program (Panlab Harvard Apparatus, Barcelona, Spain).

### 2.4. Biochemical Blood Testing

Hormonal changes and liver function were measured four weeks after administration. For biochemical blood testing, blood samples collected through a cardiac puncture of the anesthetized rats was centrifuged at 3000 rpm (1257× *g*) for 20 min (MF-300, Hanil Co., Incheon, Korea), and serum was collected and used. Aspartate aminotransferase (ALT) and alanine aminotransferase (AST) were analyzed using an AM101K kit (Asan Pharmaceutical Co., Seoul, Korea). Lipid metabolism-related indicators, such as high-density lipoprotein (HDL), low-density lipoprotein (LDL), and triglycerides, were measured using a modular analyzer (Roche Diagnostics, Basel, Switzerland).

### 2.5. Measurement of Andropause Symptoms-Related Indicators Using ELISA

The levels of andropause symptoms-related indicators, such as steroid 5 alpha-reductase 1 (SRD5A1), sex hormone-binding globulin (SHBG), dihydrotestosterone (DHT), testosterone, progesterone, luteinizing hormone (LH), and follicle-stimulating hormone (FSH), were measured by an enzyme-linked immunosorbent assay (ELISA) kit (Cusabio, Barksdale, DE, USA).

### 2.6. PTAH Staining of Muscle Tissues

To evaluate the effect of NZS extract on muscle function improvement, rat muscles aged 30 weeks old were stained with phosphotungstic acid-hematoxylin (PTAH), which were compared with normal muscles of an 8-week-old rat [28]. Muscle specimens were fixed in 10% neutral buffered formalin (NBF), and 5 um-thick paraffin sections were prepared by using a Histocut microtome (Reichert-Jung 820, Leica, Germany). Tissues subjected to deparaffinization and hydrolysis were washed in running water for 5 min and stained with 3% potassium dichromate solution for 30 min. After 10 min of reacting with 0.25% potassium permanganate solution, tissues were washed with tap water. Potassium permanganate was removed by treatment with 5% oxalic acid solution for 10 min. After washing with tap water, slides were stained with PTAH staining solution at 56 °C for 2 h, promptly dehydrated without washing, and passed through a transparent process before sealing.

### 2.7. Statistical Analysis

Statistical analysis was performed using the one-way analysis of variance (ANOVA) and Duncan multiple range test for comparison of mean values using the SPSS statistical software, where *p*-values ≤ 0.05 were considered statistically significant (SPSS 20.0 Inc., Armonk, NY, USA).

## 3. Results

### 3.1. Measure of Muscular Endurance

Male hormone deficiency causes muscle mass loss and weakness [8,29]. Therefore, the grip force and forced swim tests were performed to confirm the effect of NZS extract on the decrease in muscle mass and muscular endurance. In the grip strength test, the positive control group showed no difference from the negative control group, whereas the NZS extract-treated group had significantly increased grip strength (398.44 ± 38.63 gf, *p* < 0.05) compared to the negative control group (Figure 1A). The grip strength holding time in NZS extract-treated animals was also significantly increased (1.13 ± 0.04 s), being longer than that of the negative control group (*p* < 0.001) (Figure 1B). In addition, the forced swim test conducted to evaluate muscular endurance revealed a significantly higher immobility time in the negative control group due to the decreased body energy. No significant difference was observed in the NZS extract-treated group compared to the Testofen-treated group (positive control); however, a 8.5 ± 4.37 s (*p* < 0.01) immobility time, reduced by 11.6 times, was observed, compared to the negative control group (Figure 1C).

### 3.2. Effect of NZS Extract on Lipid Metabolism-Related Indicators

The prevalence rate of cardiovascular diseases (CVDs) increases in older men with decreased testosterone production [29]. In this study, NZS extract effectively lowered the total cholesterol level (72.0 ± 7.95 mg/dL, *p* < 0.01), compared to the negative (100.4 mg/dL) and positive (105.4 mg/dL) control groups, and also significantly decreased the triglyceride level (174.3 ± 17.83 mg/dL, *p* < 0.05) (Figure 2A,B). HDL and LDL were measured, and HDL showed no difference between treatment groups; however, LDL was significantly decreased in the Testofen- and NZS extract-treated groups compared to the negative control group (Figure 2C,D). At 11.8 ± 0.51 mg/dL LDL in the negative control group, the NZS extract-treated group showed a significant difference at 9.67 ± 0.52 mg/dL, but no difference was observed in the positive control group.

### 3.3. Effect of NZS Extract on Male Hormone Indicators

The testosterone level in the negative control group was 0.25 ± 0.07 ng/mL, whereas high levels at 0.51 ± 0.09 ng/mL and 0.47 ± 0.07 ng/mL were maintained in the positive control and NZS extract-treated groups, respectively (Figure 3A). While no significant difference was found among treatments for SRD5A1, the SHBG level was 17.44 ± 0.98 ng/mL, which is significantly lower compared to the 31.88 ± 7.13 ng/mL and 19.91 ± 2.56 ng/mL in the negative and positive control groups, respectively (Figure 3B,C).

The progesterone levels of the Testofen- and NZS extract-treated groups were 475.09 ± 109.66 ng/mL and 206.45 ± 37.47 ng/mL, respectively, which is significantly higher than that of the negative control group (Figure 3E). Therefore, the NZS treatment significantly increased progesterone levels, which is hypothesized to partially inhibit the conversion of testosterone to DHT (Figure 3D). These results suggest that NZS extracts can inhibit prostatic hyperplasia based on the preliminary evidence in our rat study.

### 3.4. Effect of NZS Extract on LH and FSH

FSH and LH are produced by the pituitary gland. LH stimulates testosterone production in Leydig cells and regulates the initiation and maintenance of spermatogenesis. FSH and LH conjointly support sperm maturation in Sertoli cells [30]; therefore, they are identified as hormones conjointly involved in spermatogenesis and other activities [31]. FSH (25.7 ± 2.56 ng/mL) and LH (40.0 ± 1.52 ng/mL) concentrations were significantly increased in the positive control group compared to the negative control group (Figure 4). Similarly, the NZS extract-treated group had increased the FSH (21.7 ± 1.70 ng/mL) and LH (31.1 ± 1.26 ng/mL) levels compared to the negative control group; however, the difference was not significant.

### 3.5. Effect of NZS Extract on ALT, AST, and PSA as Stability Indicators

Figure 5 shows the prostate-specific antigen (PSA), ALT, and AST profiles. The highest ALT expression was observed in the NZS extract-treated group; however, no significant difference was observed with the other treatments. The positive control group had the least AST expression. PSA did not change between treatment groups.

### 3.6. PTAH Staining of Muscle Tissues

For PTAH staining, the mixture of hematoxylin and PTA yielded a lake formation, staining the muscle cross striation; the remaining PTA stained the collagen fibers [32].

Therefore, normal 8-week-old rats had dense skeletal muscle fibers (myofibril), which were smooth muscle actin (α-smooth muscle actin) (Figure 6A). However, in the control group with 30-week-old rats, the distance between the muscle fibers was large, and a decrease in muscle fibers was observed, indicating the presence of a significant space between the muscle fibers (Figure 6B). In the 30-week-old rats, the decrease in muscle mass was suppressed in the groups administered Testofen^®^ (positive control) and NZS compared to the negative control group (Figure 6C,D).

## 4. Discussion

To confirm the effect of NZS on alleviating andropause syndrome, NZS extracts were administered orally to 26-week-old rats. NZS extract administration daily at 300 mg/kg body weight for four weeks reduced the cholesterol, triglyceride, and LDL levels and improved blood circulation. In addition, it alleviated andropause symptoms by reducing body fat and increasing muscle strength.

Testosterone levels reduce in men as they age, leading to a decreased muscle quality and subsequent loss of muscle mass and strength [33]. In the NZS-treated group, swimming time was longer, and muscular endurance was improved. These results indicate the possible suppressive effect of the extract on muscle mass loss; a similar trend was observed with PTAH staining, which is the basis for our hypothesis. Significant grip force and grip force time with NZS treatment resulted in significant lowering of total immobility durance, which was comparable with the Testofen group (Figure 1 and Figure 6).

According to Lee et al. (2018), the three major NZS phytochemicals are 6-methoxykaempferol-3-O-β-d-glucosyl (1‴→2″)-β-d-glucopyranoside, 6-methoxykaempferol-3-O-β-d-glucosyl (1‴→2″)-β-d-glucopyranosyl-(6⁗-caffeoyl)-7-O-β-d-glucopyranoside, and 6,4′-dimethoxykaempferol-3-O-β-d-glucosyl (1‴→2″)-β-d-glucopyranosyl-(6⁗-caffeoyl)-7-O-β-d-glucopyranoside [21]. Kaempferol exhibits antidiabetic effects by improving antioxidant activity and glucose absorption [34,35]. Varshney et al. (2019) compared serum triglyceride, cholesterol, and LDL levels after administering five types of flavonols to HFD-diabetic mice and found the most significant reduction in the kaempferol-treated group [36]. They also reported the potent anti-adipogenic activity of kaempferol using mouse pre-adipocyte 3T3-L1 cells. Therefore, NZS extract, which contains abundant 6-methoxycamperol derivatives, should improve blood circulation by reducing inflammation and lipid levels in the blood and inhibit fat synthesis by exhibiting anti-obesity effects.

Andropause symptoms include obesity, abnormal lipid metabolism, and high blood pressure. Lipid metabolism complications, such as increased LDL, decreased HDL, and increased triglycerides due to decreased testosterone, lead to cardiovascular and cerebrovascular diseases [29,37]. Higher total cholesterol levels and LDL oxidation cause atherosclerosis, which increases the incidence of CVDs, such as angina and myocardial infarction [38,39]. This study showed that NZS extract effectively lowered the total cholesterol and triglycerides compared to the control groups (Figure 2), which led to a significant decrease in the Testofen- and NZS extract-treated groups. Therefore, NZS extract might improve blood circulation by conferring beneficial changes to blood lipid metabolism.

Testosterone is the main hormone that regulates male reproductive function, is produced by Leydig cells in the testicles, and promotes spermatogenesis. In total, 70% of testosterone binds to SHBG with high affinity, and 20–30% binds to albumin, leaving approximately 3% free testosterone [23,40]. In our data, significantly higher testosterone and progesterone levels seem to inhibit the serum levels of SHBG and DHT in the Testofen and NZS treatment groups (Figure 3). Therefore, a decrease in Leydig cells reduces free testosterone, consequently inducing andropause symptoms [41]. Moreover, SHBG and testosterone conjointly induce inhibition of hormonal activity, and the SHBG levels increase as men age. Therefore, bioavailable testosterone unbound to SHBG should be high in the NZS-treated group. Compared to the positive control of the Testofen treatment group, FSH and LH, conjointly supporting sperm maturation, were not significantly improved in the NZS group, even with scoring higher than the negative control (Figure 4).

Progesterone also plays an important role in prostatic hyperplasia. Progesterone, estrogen, and gonadotropin-releasing hormone are intermediate compounds in androgen metabolism and act as antiandrogen hormones [7,29]. A previous study demonstrated that testosterone plays a role in reducing the conversion of DHT [40]. Therefore, NZS extract should inhibit prostate growth and prevent prostatic hyperplasia, as the NZS extract-treated group showed lower DHT and higher progesterone levels compared to the negative control group. PSA is a proteolytic enzyme synthesized only in prostate epithelial cells, which increases in prostatitis, prostatic hyperplasia, and prostatic infarction; therefore, it is used as a screening indicator [42]. Furthermore, according to Sutkowski et al. (1999), prostate cell proliferation is androgen-dependent and can induce receptor activation, leading to increased PSA gene transcription by binding to the PSA promoter region [43]. Therefore, elevated serum PSA levels are associated with prostate carcinoma and prostatic hyperplasia. The PSA level was significantly lowered (1.21 ng/mL) in the NZS extract-treated group by approximately 2.6 times compared to the negative control group (Figure 5). Additionally, a significant level was observed in the Testofen-treated group (0.52 ng/mL), which is not significantly different from NZS the treatment; therefore, NZS extract may prevent prostate cancer, which appears at a higher rate with age.

AST and ALT are enzymes that catalyze the trans amino acid reaction in vivo and are abundant in the liver. The progression of liver tissue destruction simultaneously increases the AST and ALT content in the blood; thus, it is widely used as a liver-damage indicator [44]. The ALS and AST levels did not change after administering NZS extract; thus, they may have no negative effect on liver function.

Cholesterol is converted to progesterone by StAR protein, CYP11A1, and 3β-HSD2, and then synthesized to testosterone by CYP17A1 and 17β-HSD3. At this time, LH and FSH produced in the pituitary gland induce the expression and activation of enzymes such as StAR protein and CYP11A1, thereby promoting the synthesis of progesterone and testosterone [45,46,47]. In this study, it is thought that the conversion of progesterone in the Testofen-treated group was promoted as the levels of LH and FSH increased. Kim et al. (2019) also reported that the Testofen-treated group showed significant increases in FSH and 3β-HSD compared to the control group, and also increased the progesterone levels, showing the same trend as in this study [24]. In addition, the level of 17β-HSD increased, resulting in increased testosterone synthesis. The NZS treatment group of this study had a significantly lower total cholesterol level compared to the Testofen treatment group, and the final total testosterone did not show a significant difference compared to the Testofen group. It is thought that the NZS treatment promoted the expression of the StAR protein and CYP11A1, which convert cholesterol into pregnenolone(progesterone precursor). Furthermore, the level of progesterone did not reach the increase seen in the Testofen-treated group, which is expected because the expression of CYP17A1 and 17β-HSD3, which converts progesterone into testosterone, increased. Therefore, in order to prove the hypothesis, it is suggested to investigate the mRNA expression and activation of testosterone synthesis-related enzymes (StAR protein, CYP11A1, 3β-HSD, CYP17A1, 17β-HSD3, etc.).

## 5. Conclusions

The ethanol extract of NZS improved the performance and facilitated lipid metabolism. In addition, the DHT and SHBG levels were decreased, and the progesterone levels were significantly increased, indicating a positive effect on testosterone synthesis. Based on our evidence in this aging male rat model, NZS can alleviate andropause symptoms and inhibit prostatic hyperplasia, which can be potentially used as a natural functional material.

## Figures and Tables

**Figure 1 nutrients-14-04572-f001:**
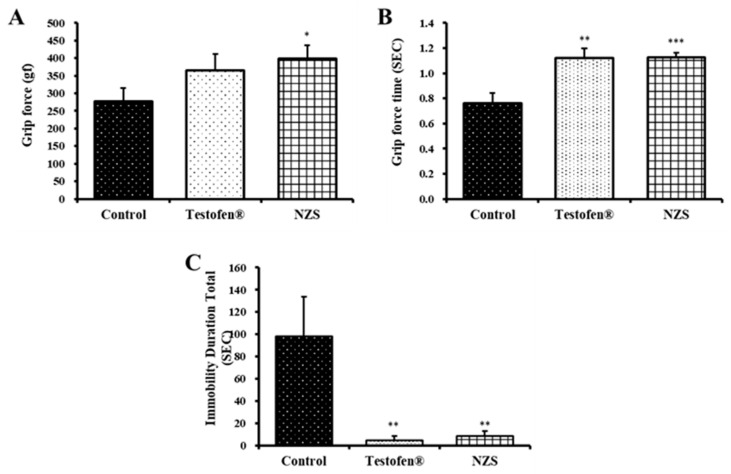
Effects of New Zealand spinach (NZS) extract on grip strength. (**A**) Force value indicates the pulling force in gf (peak force). (**B**) Resistance time during the test. (**C**) Immobility time for 12 min in forced swimming. Control, no administration of NZS extract; Testofen as a positive control; and NZS (300 mg/kg Testofen and NZS extract treatment group). Data are expressed as means ± standard error (n = 5). Symbols of *, **, and *** denote significant differences compared to aged rat (control) at *p* < 0.05, *p* < 0.01, and *p* < 0.001 levels, respectively.

**Figure 2 nutrients-14-04572-f002:**
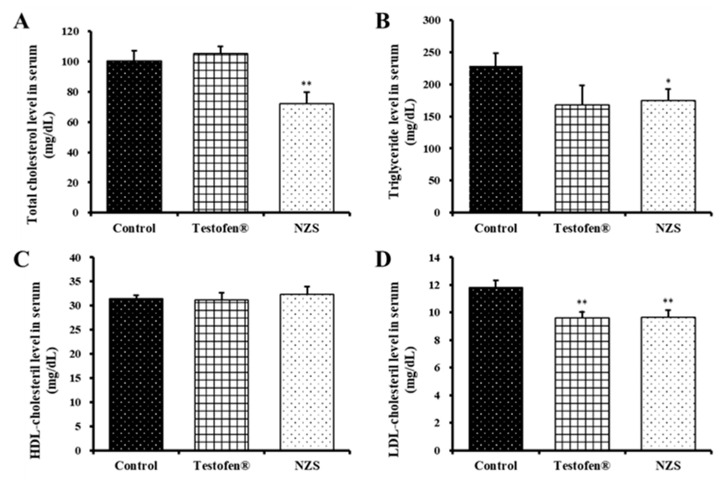
Comparative analysis of the serum levels of total cholesterol (**A**), triglyceride (**B**), high-density lipoprotein (HDL) (**C**), and low-density lipoprotein (LDL) (**D**) after New Zealand spinach (NZS) extract administration for four weeks. Data are expressed as means ± standard error (n = 5). Symbols of * and ** denote significant differences compared to aged rat (control) at *p* < 0.05 and *p* < 0.01 levels, respectively.

**Figure 3 nutrients-14-04572-f003:**
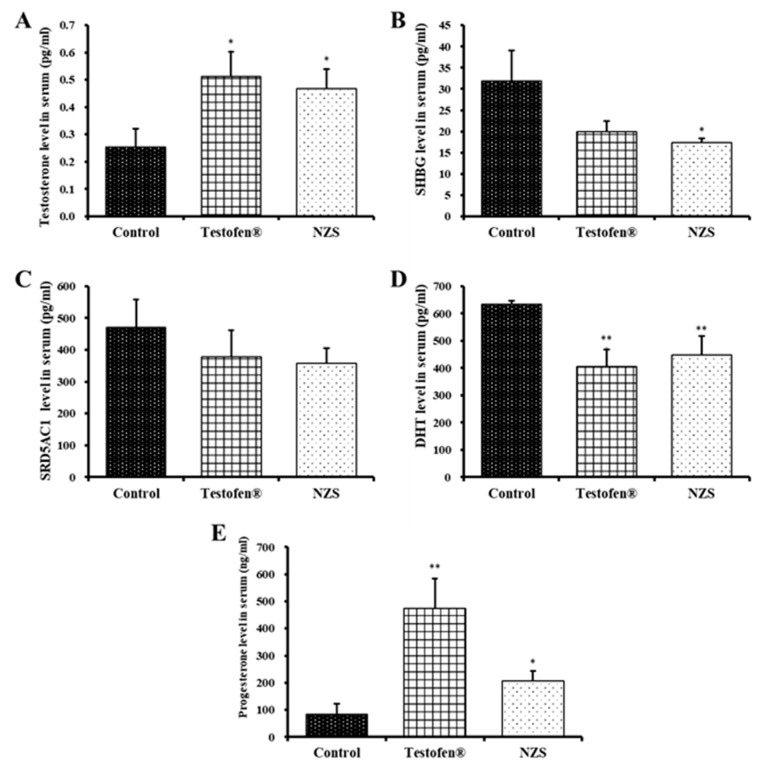
Comparative analysis of serum levels of testosterone (**A**), sex hormone-binding globulin (SHBG) (**B**), steroid 5 alpha-reductase 1 (SRD5A1) (**C**), dihydrotestosterone (DHT) (**D**), and progesterone (**E**) after NZS extract administration for four weeks. Data are expressed as means ± standard error (n = 5). Symbols of * and ** denote significant differences compared to aged rat (control) at *p* < 0.05 and *p* < 0.01 levels, respectively.

**Figure 4 nutrients-14-04572-f004:**
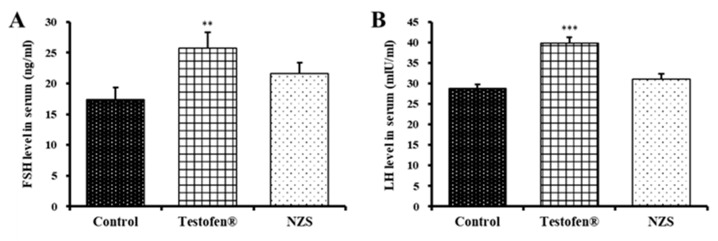
Comparative analysis of serum levels of follicle-stimulating hormone (FSH) (**A**) and luteinizing hormone (LH) (**B**) after NZS extract administration for four weeks. Data are expressed as means ± standard error (n = 5). Symbols of **, and *** denote significant differences compared to aged rat (control) at *p* < 0.01, and *p* < 0.001 levels, respectively.

**Figure 5 nutrients-14-04572-f005:**
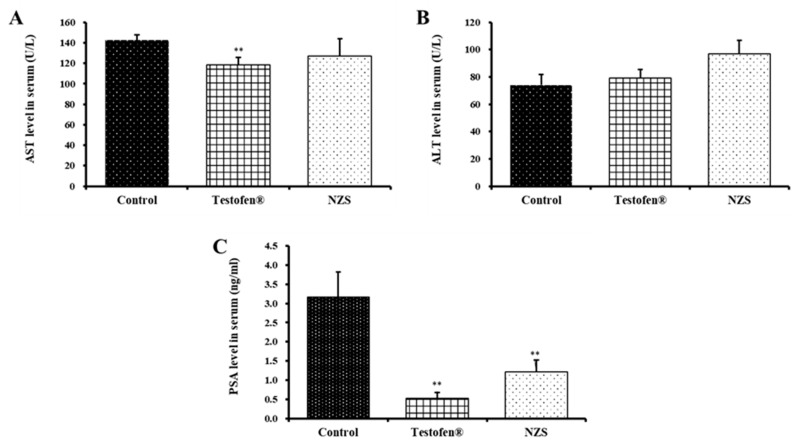
Comparative analysis of serum levels of aspartate aminotransferase (ALT) (**A**), alanine aminotransferase (AST) (**B**), and prostate-specific antigen (PSA) (**C**) after NZS extract administration for four weeks. Data are expressed as means ± standard error (n = 5). A symbol of ** denotes significant difference compared to aged rat (control) at *p* < 0.01 level.

**Figure 6 nutrients-14-04572-f006:**
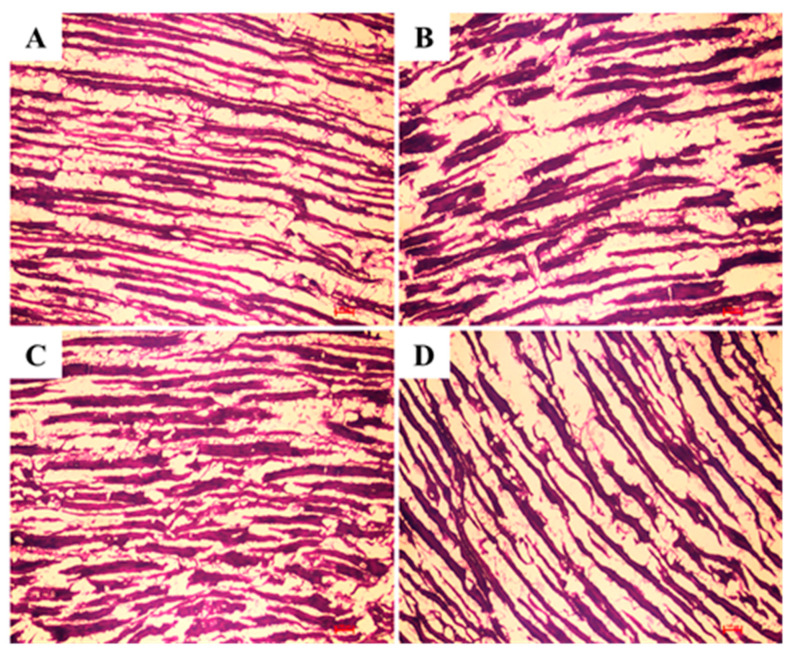
Muscle sections were prepared to form different groups of animals and subjected to phosphotungstic acid-hematoxylin (PTAH) staining. Normal Sprague Dawley (SD) rat (**A**); aged SD rat (**B**); aged SD rat treated with Testofen^®^ (**C**); aged SD rat with NZS extracts (**D**).

## Data Availability

Data is contained within the article.

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
