# Peer review of "Effect of Tetragonia tetragonoides (Pall.) Kuntze Extract on Andropause Symptoms"

_nutrients, 2022, doi:10.3390/nu14214572_

Round 1

Reviewer 1 Report

The issue of andropause is important to most humans, so any insights into how to arrest some of the symptoms associated with it during aging, along with further insights into what changes would most benefit, warrants such studies. For this reason, the authors are to be commended for taking on this study and reporting the results. 

Lines 43 – 54. Suggest the authors remove this section since the variability of these studies in terms animal species, dose administration, lack of replication, etc., and move straight to the discussion starting on line 55 on T. tetragonoides. 

Lines 55-56. The authors should include information on other names for this halophyte, including everbearing spinach, and perpetual spinach. Explain why it is different in terms of leaf texture and succulent characteristic from other commonly available spinach found in marketplaces. Some added information should mention it relatively high salt and oxalate acid content (reported at 506-981 mg/100 g), the latter of importance in combination with calcium, to prevent oxalate stone-forming in individuals monitoring oxalate intake to reduce the risk of kidney stones. Steaming and boiling can reduce oxalate content, with the latter most effective. The paper by Grazyna Jaworska (Food Chem, 89(2): 235-242, 2005) points out that water-soluble oxalates in total oxalates is 59% for spinach compared to 81% for NZ spinach. However, NZ spinach is a rich source of vitamin K, which provides multiple benefits in supporting health. However, cooked spinach has much higher levels of vitamin K than saw spinach, which would be important to know for individuals taking warfarin, as it can affect the drugs effectiveness. Hence, given the method of extraction mentioned at line 83, what is the effect on vitamin K content of the extract since it is obtained via “hot water extraction”?

Given the above comments, how does the extract address these concerns or issues? 

Some mention is needed in the introduction that the decline in testosterone with age is related to a decrease in testicular and hypothalamic function due to a decrease in gonadotropin-releasing hormone resulting in a reduction in the production of luteinizing hormone by the pituitary gland. This background provides the reader some reference to the author’s reported findings. For the extract to be of value it would have to improve symptoms and restore age-related alterations in physiology and biochemistry seen in andropause. 

Line 85, details are needed on how freeze-drying of the extract was performed, including instrumentation. What analytical chemistry was performed, and its results, to confirm the leaves were T. tetragonoides? 

Lines 88-94. How many animals were housed per cage, what type of cage, and bedding? Indicate the number of animals purchased and how many were used in the study. How was assignment to each group performed? 

Line 95, provide the name of the manufacturer and source (city/state) for the control extract. How is the comparison between the brand supplement, Testofen, that served as the control comparable in terms of what has been reported is its mechanisms of action? For example, Kim reported in the J Korean Soc Food Sci Nutr (2020) that the extract of Trigonella foenun-graecum seed (Testofen) lowered levels of aromatase and 5-a reductase in vivo inducing stimulation of follicle stimulating hormone (that activate Leydig cells). This lowering of enzymes levels that degrade testosterone resulted in an increase in testosterone synthesis. Hence, how would using this control extract be appropriate unless the same observed effects reported for it are also measured in the experimental group? Were there significant changes seen or equivalency observed when comparing the two groups for the level of aromatase and 5-a-reductase? How do the authors justify comparing the positive control with the experimental extract? Was the 300 mg/kg bw adequate for comparison purposes? No mention is made whether the 300 mg dose was administered in the rat. What happened to the animals at the end of the study? While the authors report they followed animal care guidelines, this still requires additional information on how the animals were handled since changes in personnel that handle the animals, for example, can affect hormone levels, and stress levels. The more information the authors provide the easier it will be to replicate this study. 

Based on the authors selection of Testofen as a control, did the literature reveal any studies performed on it that included the measures selected for their study for comparison? 

Line 182. When reaching conclusions on a potential benefit point out that the hypothesized benefit to “inhibit prostatic hyperplasia” is based on preliminary evidence in the rat. 

The Discussion and Conclusion. 

While the authors illustrate the results in a series of bar graphs starting at line 158, the discussion and conclusion is devoid of mention of the positive control, Testofen. Highlighting in the discussion, and certainly within the conclusion, the key statistical differences in outcomes between the experimental and positive control groups would seem useful for the reader to see summarized, rather than return to each graph to figure out differences, given the need to read text on the y-axis that is printed vertically.

Guidance on what studies are needed in light of the author's findings should be included in the discussion or conclusion, as well as any plans the authors have made to continue performing research on NZ spinach. 

Reviewer 2 Report

In this study, Tetragonia tetragonoides (Pall.) Kuntze extract was used in mice to carry out its anti-aging, especially the symptoms of male andropause symptoms. This research can provide basis for food therapy or the development of health products. However, this study has the following defects:

1. Rats may be more suitable for this study.

2. There are at least 10 mice in each group.

3. There are too few groups of mice, only negative group, positive group and extract group. The extract group should have at least three groups: high, medium and low.

4. The main components of the extract are unknown.

5. The active ingredients of the extract are not clear.

6. The possible mechanism of action of the extract was not analyzed. The research results only include physiological data, which is obviously not enough.

Reviewer 3 Report

In my opinion, the article presented to me for review raises an important topic. The conducted research is innovative and uses advanced analytical methods.

The experiment was properly planned. The introduction is well written, not too long, and provides the appropriate background.

Objective research is valuable, well presented and statistically analyzed.

The discussion of the results is based on the latest references.

The conclusions are supported by the conclusions.

The language of the manuscript is comprehensible and legible.

In my opinion, the work deserves a publication in Nutrients.

Round 2

Reviewer 1 Report

Excellent additions were added in the revision that will help the readers understand the mechanism by which the abstract worked. Note the misspelling at line 332 of the revision. 

Reviewer 2 Report

The author did make some improvements, however, problems still exist. In the absence of any clarification of the material basis and mechanism of action, I still think it is not suitable to publish it in top journals.